# Evolutionary Diversity Optimization with Clustering-based Selection for Reinforcement Learning

**Yutong Wang**[*]**, Ke Xue**[*] **and Chao Qian**[†]
State Key Laboratory for Novel Software Technology,
Nanjing University, Nanjing 210023, China
`{wangyt, xuek, qianc}@lamda.nju.edu.cn`

## Abstract

Reinforcement Learning (RL) has achieved significant successes, which aims to obtain a single policy maximizing the expected cumulative rewards for a given task. However, in many real-world scenarios, e.g., navigating in complex environments and controlling robots, one may need to find a set of policies having both high rewards and diverse behaviors, which can bring better exploration and robust few-shot adaptation. Recently, some methods have been developed by using evolutionary techniques, including iterative reproduction and selection of policies. However, due to the inefficient selection mechanisms, these methods cannot fully guarantee both high quality and diversity. In this paper, we propose EDO-CS, a new Evolutionary Diversity Optimization algorithm with Clustering-based Selection. In each iteration, the policies are divided into several clusters based on their behaviors, and a high-quality policy is selected from each cluster for reproduction. EDO-CS also adaptively balances the importance between quality and diversity in the reproduction process. Experiments on various (i.e., deceptive and multi-modal) continuous control tasks, show the superior performance of EDO-CS over previous methods, i.e., EDO-CS can achieve a set of policies with both high quality and diversity efficiently while previous methods cannot.

## 1 Introduction

Reinforcement Learning (RL) is an effective method for training agents to make decisions in a given environment, which is often to obtain a policy maximizing the expected cumulative rewards (Li, 2017; Sutton & Barto, 2018). As RL is applied to more real-world scenarios, training a single policy is insufficient to handle complex problems, and we may need to find a set of policies with both high quality (i.e., rewards) and diverse behaviors. For example, when navigating in complex environments (Zhu et al., 2017; Mirowski et al., 2017), there are usually deceptive rewards trapping a single policy into the local optima. Maintaining a set of policies allows data to be collected with diverse behaviors and may lead to better exploration in these environments. When controlling robots, it is fragile to have one single policy (Cully et al., 2015). In contrast, maintaining a set of policies can improve robustness, e.g., enable real robots to recover quickly from joint damage (Cully et al., 2015), and also be helpful for few-shot adaption when facing unseen tasks (Kumar et al., 2020). General deep RL algorithms mostly focus on achieving high rewards, restricting their ability to generate diverse behaviors.

---

[*]Equal contribution
[†]Corresponding author

Evolutionary Algorithms (EAs) are general-purpose heuristic optimization algorithms that maintain a population (i.e., a set of solutions), and simulate the natural evolution process with iterative reproduction and selection (Bäck, 1996). Quality-Diversity (QD) algorithms (Pugh et al., 2016; Cully & Demiris, 2018) are a specific type of EAs that aim to return a set of high-quality solutions with diverse behaviors in a single run. They have been naturally applied to RL in complex environments, generating the corresponding RL algorithms NSR-ES (Conti et al., 2018), ME-ES (Colas et al., 2020), DvD-ES (Parker-Holder et al., 2020) and QD-RL (Cideron et al., 2020). Note that classical QD algorithms like NSLC (Lehman & Stanley, 2011a) or MAP-Elites (Mouret & Clune, 2015; Cully et al., 2015) often employ a Genetic Algorithm (GA) as the underlying optimizer. However, as RL tasks are often high-dimensional, these methods have replaced GA with Evolution Strategies (ES) presented by Salimans et al. (2017), which has achieved performance comparable to state-of-the-art deep RL algorithms on high-dimensional control tasks.

NSR-ES (Conti et al., 2018) maintains a population of ES agents and introduces Novelty Search (NS) (Lehman & Stanley, 2011b) to improve the performance on RL tasks with sparse or deceptive rewards. That is, it adds a novelty term (representing behavior diversity) to the objective function of ES agents. In each iteration, NSR-ES selects a policy from the population for reproduction, where the probability of selecting a particular policy is proportional to its novelty score.

In contrast to NSR-ES (Conti et al., 2018), DvD-ES (Parker-Holder et al., 2020) optimizes all policies of a population simultaneously, by maximizing the sum of their rewards and a population-wide diversity criterion (i.e., the volume between the behaviors of the policies). Though DvD-ES can maintain a diverse population, it ignores the previously generated policies, leading to "cycling" (i.e., a phenomenon where the population keeps moving alternatively between two areas) and thus inefficient performance.

It has been shown (Cully & Demiris, 2018) that keeping an additional archive (i.e., a subset of policies generated-so-far) and selecting policies from the archive instead of the population for reproduction can be beneficial. There are different ways to select policies from the archive. ME-ES (Colas et al., 2020) uses an alternating selection mechanism, which selects several policies with the best quality in one iteration and with the best diversity in the next iteration. QD-RL (Cideron et al., 2020) uses Pareto-based selection (Deb et al., 2002) by considering both quality and diversity in one iteration, and achieves better performance. That is, the Pareto front is first calculated with respect to quality and diversity, and then the policies are selected from the Pareto front based on the crowding distance.

Although Pareto-based selection can guarantee the selected policies uniformly distributed over the Pareto front, these policies may be still similar in the behavior space, limiting the performance of QD-RL. For example, Figure 1(a) shows the heat map of a synthetic function $f(x, y)$ in the behavior space, where the color corresponds to the function value, which we call quality here. The goal is to find the four optimal solutions distributed in the center of four regions of the behavior space. The points represent the solutions in the current archive, from which we need to select some solutions for reproduction. QD-RL first calculates the diversity of each solution (i.e., the average Euclidean distance between this solution and its $k$-nearest neighbors) in the archive, and then gets the Pareto front with respect to quality and diversity, as shown in Figure 1(b). By Pareto-based selection, the blue points together with the red star will be selected, which are uniformly distributed over the Pareto front, but concentrated in the lower right corner of the behavior space.

In this paper, we propose an Evolutionary Diversity Optimization algorithm with Clustering-based Selection (EDO-CS). In each iteration, the policies in the archive are divided into several clusters based on their behaviors, and a high-quality policy is selected from each cluster for reproduction. This selection mechanism can naturally guarantee the diversity of the selected policies in the behavior space. For example, the red points and red star in Figure 1(a) will be selected by clustering-based selection, which are distributed in the four different areas of the behavioral space.

To examine the performance of EDO-CS, we conduct experiments on a variety of continuous control tasks from *OpenAI Gym* library (Brockman et al., 2016). Firstly, we show that EDO-CS can solve navigating

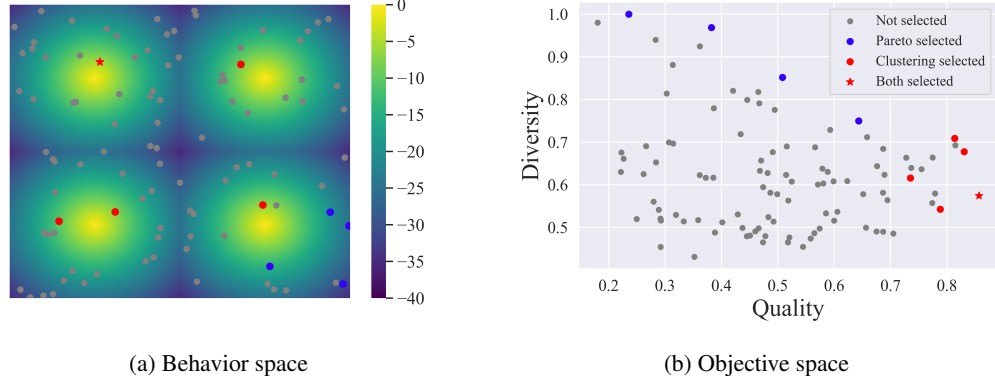

(a) Behavior space            (b) Objective space

Figure 1: The selected solutions by Pareto-based selection and clustering-based selection on a synthetic problem. Figure 1(a) shows the behavior space, where the color of the heat map shows the quality of the solution. Figure 1(b) shows the quality and diversity of each solution.

problems in environments with deceptive rewards. Then, we test it in multi-modal tasks, showing that EDO-CS can efficiently find optimal policies with diverse behaviors. In the standard MuJoCo environments that do not require much diversity, we show that EDO-CS can achieve even better performance than Vanilla ES that focuses on the quality. We also test the influence of the hyper-parameters of EDO-CS in these environments.

The remainder of this paper is organized as follows. We first introduce some definitions and notations used in our paper in Section 2. After that, we propose our method in Section 3. The results and analyses of experiments are shown in Section 4. Finally, we conclude this work in Section 5.

## 2   BACKGROUND

We consider the setting of a fully observable Markov Decision Process (MDP) described as a tuple $(\mathcal{S}, \mathcal{A}, P, R, \gamma)$, where $\mathcal{S}$ is the state space, $\mathcal{A}$ is the action space, $P(s_{t+1}|s_t, a_t)$ gives the probability of transiting to state $s_{t+1}$ after taking action $a_t$ at state $s_t$, $r_t = R(s_t, a_t, s_{t+1})$ is the reward obtained by transiting to state $s_{t+1}$ after taking action $a_t$ at state $s_t$, and $\gamma$ is the discount factor.

A policy $\pi_{\boldsymbol{\theta}} : \mathcal{S} \rightarrow \mathcal{A}$ is a mapping from state space $\mathcal{S}$ to action space $\mathcal{A}$, paremeterized by $\boldsymbol{\theta}$, which can be either deterministic or randomized. The goal of general RL algorithms is to optimize the parameters $\boldsymbol{\theta}$ such that $\pi_{\boldsymbol{\theta}}$ maximizes the expectation of the cumulative rewards $R(\tau) = \sum_{t=0}^{T} \gamma^t r_t$, where $\tau$ is the trajectory over a horizon $T$. The objective function can be written as $J(\boldsymbol{\theta}) = \mathbb{E}_{\tau \sim \pi_{\boldsymbol{\theta}}}\left[R(\tau)\right]$.

To make better exploration and have more robustness, QD algorithms seek to find a set of policies with both high rewards and diverse behaviors (Pugh et al., 2016; Cully & Demiris, 2018). The behavior characterization $b(\pi_{\boldsymbol{\theta}})$ of a policy $\pi_{\boldsymbol{\theta}}$ is usually domain-dependent. For example, it can be the coordinates of the final location $(x, y)$ of the robot in the robot locomotion problem, and the corresponding behavior space is $\mathbb{R}^2$. It can also be defined as the actions taken by policies, leading to a high-dimensional behavior space. Given an archive $A$ of policies, the diversity of a particular policy $\pi_{\boldsymbol{\theta}}$ is computed as the average Euclidean distance between $b(\pi_{\boldsymbol{\theta}})$ and $A'_{\pi_{\boldsymbol{\theta}}, k}$, the set of $k$-nearest neighbors of $\pi_{\boldsymbol{\theta}}$ in the archive $A$. That is:

$$Div(\boldsymbol{\theta}) = \frac{1}{k} \sum_{\pi'_{\boldsymbol{\theta}} \in A'_{\pi_{\boldsymbol{\theta}}, k}} \|b(\pi_{\boldsymbol{\theta}}) - b(\pi'_{\boldsymbol{\theta}})\|_2. \tag{1}$$

Then, the objective function to be maximized can be expressed as:

$$J(\boldsymbol{\theta}) = (1 - \lambda) \cdot \mathbb{E}_{\tau \sim \pi_{\boldsymbol{\theta}}} [R(\tau)] + \lambda \cdot Div(\boldsymbol{\theta}), \tag{2}$$

where $\lambda$ controls the trade-off between the quality (i.e., rewards) and diversity.

## 3 THE EDO-CS ALGORITHM

In this section, we introduce the proposed algorithm, Evolutionary Diversity Optimization with Clustering-based Selection (EDO-CS). We first give an overview of EDO-CS in Section 3.1. Then, we introduce the selection and reproduction mechanisms in Sections 3.2 and 3.3, respectively. Section 3.4 shows how to balance the quality and diversity. Finally, we compare EDO-CS with other QD algorithms in Section 3.5.

### 3.1 OVERVIEW

A general flow of EDO-CS is shown in Algorithm 1. It first initializes an archive $A$ with a maximum capacity of $l$, which contains the policies generated-so-far together with their corresponding cumulative rewards and behaviors; and a bandit $B$, which is used to sample the weight of diversity (i.e., $\lambda$ in Eq. (2)) during the reproduction process. Then, it adds $l$ randomly generated policies into the archive $A$ (lines 1–5), and tries to improve the policies in $A$ iteratively (lines 7–20). In each iteration, EDO-CS employs $K$-means (Lloyd, 1982) to partition the policies in the archive into $K$ clusters based on their behaviors (line 8), and selects one specific policy from each cluster (line 9). After getting $K$ policies, EDO-CS samples the value of $\lambda$ in the objective function for each policy from the bandit $B$, and uses the ES optimizer to update the selected policies in parallel for $T'$ iterations (lines 11–15). The archive and bandit are updated in line 18 after evaluating all the generated offspring policies. To maintain the diversity of the archive, we avoid adding policies with similar behaviors (Cully & Mouret, 2013). To be specific, whenever we want to add a new policy into the archive, we first select the policy with the most similar behavior from the archive. If their behavior distance is greater than a threshold, we add the new policy into the archive. Otherwise, we compare their rewards and keep the policy with a higher reward. Note that if the archive size exceeds the capacity $l$ after adding a new policy, the oldest policy in the archive will be deleted. EDO-CS repeats the above process until the number of iterations reaches $T$.

### 3.2 CLUSTERING-BASED SELECTION

The selection process in each iteration of EDO-CS tries to select a set of high-quality policies with diverse behaviors from the archive. Here, we employ a natural and efficient way, i.e., clustering-based selection, which first clusters the policies in the archive and then selects a high-quality policy from each cluster. Clustering is an unsupervised algorithm for finding natural groups of a data set, such that the data in the same group (i.e., a cluster) are similar while those in different groups are not similar. For a given data set $D = \{\boldsymbol{x_1}, \boldsymbol{x_2}, \cdots, \boldsymbol{x_n}\}$, a clustering algorithm divides $D$ into $K$ clusters $C = \{C_1, C_2, \cdots, C_K\}$ where $\forall i \neq j, C_i \cap C_j = \emptyset$ and $D = \cup_{i=1}^{K} C_i$. $K$-means (Lloyd, 1982) is one of the most popular clustering algorithms, which aims to minimize within-cluster variances, i.e.,

$$\arg \min_{C} \sum_{i=1}^{K} \sum_{\boldsymbol{x} \in C_i} \|\boldsymbol{x} - \boldsymbol{\mu}_i\|_2^2, \tag{3}$$

where $\boldsymbol{\mu}_i$ is the mean of points in $C_i$.

In the selection process, we first use $K$-means to cluster the policies in the archive based on their behaviors, which means $\boldsymbol{x} = b(\pi_{\boldsymbol{\theta}})$ in Eq. (3). Then, we select one specific policy $\pi_{\boldsymbol{\theta}_i}$ from each cluster $C_i$. More specifically, we first select the policy with the highest quality from the archive, and the corresponding cluster

---

**Algorithm 1:** EDO-CS

---

**Input:** number $K$ of selected policies, number $T$ of total iterations, number $T'$ of updating
      iterations, behavior characterization $b(\pi_{\boldsymbol{\theta}})$, archive size $l$, archive $A$, bandit $B$

**Output:** archive $A$

    // Warm up

1 **for** $j = 1 : l$ **do**

2     Randomly generate policy $\pi_{\boldsymbol{\theta}_j}$;

3     Get cumulative rewards $R$ and behavior $b(\pi_{\boldsymbol{\theta}_j})$ by evaluating the policy $\pi_{\boldsymbol{\theta}_j}$;

4     Add $(\pi_{\boldsymbol{\theta}_j}, R, b(\pi_{\boldsymbol{\theta}_j}))$ into archive $A$

5 **end**

6 $t = 0$;

7 **while** $t < T$ **do**

    // Selection

8     Use $K$-means to divide the policies in archive $A$ into $K$ clusters $\{C_k\}_{k=1}^K$;

9     Select $K$ policies $\{\pi_{\boldsymbol{\theta}_k}\}_{k=1}^K$, each one from a cluster;

    // Reproduction

10     **for** $k = 1 : K$ **do**

        // Update in parallel

11         Sample $\lambda_k$ from the bandit $B$;

12         **for** $i = 1 : T'$ **do**

13             Set the objective function $J(\boldsymbol{\theta}_k) = (1 - \lambda_k)\mathbb{E}_{\tau \sim \pi_{\boldsymbol{\theta}_k}}[R(\tau)] + \lambda_k Div(\boldsymbol{\theta}_k)$;

14             Use ES to update $\boldsymbol{\theta}_k$ as Eq. (6)

15         **end**

16         Get cumulative rewards $R$ and behavior $b(\pi_{\boldsymbol{\theta}_k'})$ by evaluating the updated policy $\pi_{\boldsymbol{\theta}_k'}$

17     **end**

18     Update archive $A$ and bandit $B$;

19     $t = t + T'$

20 **end**

---

is denoted as $C_{i^*}$. Then, for each other cluster $C_i \neq C_{i^*}$, we find its $M$ highest-quality policies and select one uniformly at random. Policies in the same cluster have similar behaviors, and each selected policy can be viewed as a representative of its cluster, which has the potential to reproduce a high-quality policy with this type of behavior. Thus, by this clustering-based selection process, we can select good policies from different areas of the behavior space for reproduction.

Note that some previous works (Vassiliades et al., 2017; 2018) have also introduced clustering into QD algorithms to ensure diversity. But the usages are quite different. Previous works use the grid-based container and employ clustering to partition a high-dimensional behavior space into well-spread geometric regions before running the iterative evolutionary process, while our EDO-CS employs clustering to select a set of high-quality solutions with diverse behaviors from the archive in each iteration.

### 3.3 ES-BASED REPRODUCTION

To reproduce offspring solutions from the selected solutions, we use ES as the optimizer. Inspired by natural evolution, ES is a broad class of population-based black-box optimization algorithms, while here we use the version introduced in Salimans et al. (2017), which has achieved performance comparable to state-of-the-art deep RL algorithms on high-dimensional control tasks. ES represents a population of parameters $\boldsymbol{\theta}$ by a

distribution $p_\phi(\boldsymbol{\theta})$, parameterized by $\phi$, and seeks to optimize the expected objective value $\mathbb{E}_{\boldsymbol{\theta} \sim p_\phi}[J(\boldsymbol{\theta})]$, where $J(\boldsymbol{\theta})$ is an objective function to measure the goodness of parameters $\boldsymbol{\theta}$. Given the population distribution $p_{\phi_t}$, parameters $\boldsymbol{\theta}_t^i \sim \mathcal{N}(\boldsymbol{\theta}_t, \sigma^2 \boldsymbol{I})$ are sampled and evaluated. Using the log-likelihood trick, $\boldsymbol{\theta}_t$ is updated using an estimation of the gradient of expected reward:

$$\nabla_\phi \mathbb{E}_{\boldsymbol{\theta} \sim p_\phi}[J(\boldsymbol{\theta})] \approx \frac{1}{n} \sum_{i=1}^n J(\boldsymbol{\theta}_t^i) \nabla_\phi \log p_\phi(\boldsymbol{\theta}_t^i), \tag{4}$$

where $n$ is the number of samples evaluated per generation. In practice, any $\boldsymbol{\theta}_t^i$ can be decomposed as $\boldsymbol{\theta}_t^i = \boldsymbol{\theta}_t + \sigma \boldsymbol{\epsilon}^i$, where $\sigma > 0$ and $\boldsymbol{\epsilon}^i \sim \mathcal{N}(0, \boldsymbol{I})$. Thus, the gradient estimation in Eq. (4) becomes

$$\nabla_{\boldsymbol{\theta}_t} \mathbb{E}_{\boldsymbol{\epsilon} \sim \mathcal{N}(0, \boldsymbol{I})}[J(\boldsymbol{\theta}_t + \sigma \boldsymbol{\epsilon})] \approx \frac{1}{n\sigma} \sum_{i=1}^n J(\boldsymbol{\theta}_t^i) \boldsymbol{\epsilon}^i. \tag{5}$$

Based on the above gradient estimation, the parameter $\boldsymbol{\theta}_t$ will be updated as

$$\boldsymbol{\theta}_{t+1} = \boldsymbol{\theta}_t + \frac{\eta}{n\sigma} \sum_{i=1}^n J(\boldsymbol{\theta}_t^i) \boldsymbol{\epsilon}^i, \tag{6}$$

where $\eta$ is the learning rate of the ES optimizer. In this paper, we set $J(\boldsymbol{\theta})$ as Eq. (2), i.e., a weighted sum of quality and diversity. But as different values of the hyper-parameter $\lambda$ in Eq. (2) may be required for different tasks as well as different stages of the same task, we self-adjust it based on multi-armed bandit, which will be presented in the next subsection.

## 3.4 ADAPTIVE BALANCE BETWEEN QUALITY AND DIVERSITY

In Eq. (2), the reward and diversity correspond to exploitation and exploration, respectively, implying that the hyper-parameter $\lambda$ controls the trade-off between exploitation and exploration. Here, we model the problem of setting an appropriate $\lambda$ as a multi-armed bandit problem (Vermorel & Mohri, 2005) by treating each alternative $\lambda$ as an arm of the bandit. Let $\Lambda = \{\lambda^{(1)}, \lambda^{(2)}, \dots, \lambda^{(m)}\}$ denote the set of alternative arms. For each arm $\lambda^{(i)}$, the gain of rewards after updating a policy with $\lambda^{(i)}$ is used as its reward. Note that to use the multi-armed bandit model, we have made a stationary assumption, i.e., disregarding the change of the reward distribution of each arm in the process of optimization.

The Upper Confidence Bound (UCB) algorithm (Auer, 2002) is used to select a specific arm in each iteration. That is, the arm $\lambda_t$ in iteration $t$ is chosen as

$$\lambda_t = \arg\max_{\lambda^{(i)}} \mu_t(\lambda^{(i)}) + c\sqrt{\frac{\ln t}{N_t(\lambda^{(i)})}}, \tag{7}$$

where $c > 0$, $\mu_t(\lambda^{(i)})$ is the estimated mean reward of arm $\lambda^{(i)}$, and $N_t(\lambda^{(i)})$ is the number of times we have selected $\lambda^{(i)}$. Note that $\mu_t(\lambda^{(i)})$ reflects the current knowledge of the algorithm in a condensed form and guides further exploitation, while $\sqrt{\frac{\ln t}{N_t(\lambda^{(i)})}}$ (i.e., the width of the confidence bound) reflects the uncertainty of the algorithm's knowledge and guides further exploration. The ablation study of the effectiveness of multi-armed bandit can be found in Appendix A.2.

## 3.5 COMPARISON OF QD ALGORITHMS

Finally, we compare EDO-CS with other related QD algorithms from the perspective of EAs, as shown in Table 1. The second column gives the selection strategies, and the third column shows whether the quality or diversity is considered in the objective function for reproducing offspring solutions. In the column of "EAs type", the algorithm maintains $\mu$ solutions in the population, and generates $\lambda$ offspring solutions in each iteration, where $(\mu + \lambda)$ and $(\mu, \lambda)$ denote that the parent solutions will and will not be selected, respectively. The last column indicates whether the algorithm selects the solutions from the archive.

Table 1: A review of QD algorithms from the perspective of EAs.

| Method | Selection | Reproduction | EAs type | From archive |
|---|---|---|---|---|
| Vanilla ES | The only parent solution | Quality | $(1, 1)$ | × |
| NSR-ES | Probabilistic selection | Quality and diversity | $(K, 1)$ | × |
| CVT-ES | Uniform selection | Quality and diversity | $(K + K)$ | ✓ |
| ME-ES | Biased selection | Quality or diversity | $(K + K)$ | ✓ |
| DvD-ES | All parent solutions | Quality and diversity | $(K, K)$ | × |
| QD-RL | Pareto-based selection | Quality or diversity | $(K + K)$ | ✓ |
| **EDO-CS** | Clustering-based selection | Quality and diversity | $(K + K)$ | ✓ |

## 4    EXPERIMENT

To examine the performance of EDO-CS, we conduct experiments on a variety of continuous control tasks from the *OpenAI Gym* library (Brockman et al., 2016), including deceptive, multi-modal and standard MuJoCo environments. We compare EDO-CS against Vanilla ES (Salimans et al., 2017), NSR-ES (Conti et al., 2018), CVT-MAP-Elites (Vassiliades et al., 2018), ME-ES (Colas et al., 2020), DvD-ES (Parker-Holder et al., 2020) and QD-RL (Cideron et al., 2020). For a fair comparison, we use the ES optimizer with the same hyper-parameters in all algorithms, and CVT-MAP-Elites is denoted as CVT-ES accordingly. The population size $K$ of these algorithms is always set to 5. For EDO-CS, the number $M$ of candidates for selection in each cluster is set to 2, and the arms of the bandit are $\{\lambda^{(1)} = 0, \lambda^{(2)} = 0.5\}$. Other parameter settings can be found in Appendix A.1. To have a sample-wise fair comparison, each algorithm uses the same number (i.e., 3000) of generations of ES optimizer. Note that each of the 600 iterations in the $x$-axis of all figures corresponds to five generations. Thus, for Vanilla ES and NSR-ES which reproduce only one policy in each iteration, one unit in the $x$-axis actually corresponds to their five iterations. Other algorithms reproduce five policies in each iteration, and thus one unit in the $x$-axis just corresponds to their one iteration. We report the mean and standard deviations across six identical seeds (2016 - 2021) for all algorithms and all tasks.

### 4.1    DECEPTIVE ENVIRONMENTS

First, we consider an environment with deceptive rewards, which is modified from the standard *Ant-v2* environment. As shown in Figure 2(a), a U-shape wall is in front of the ant, separating the ant from the target, i.e., green cuboid. The reward is a mixture of distance away from the target and costs for joint torques, encouraging the ant to walk to the target as fast as possible. Obviously, following the gradient of only the expected cumulative rewards will make the ant walk forward directly and get trapped by the wall. Here we use the final location $(x, y)$ of the ant as the behavior characterization.

Figure 2(b) shows the rewards of the best-performing policy in the current archive in each iteration. As expected, both Vanilla ES and DvD-ES fail to get past the wall, because they do not use an archive to record previously explored policies, and thus cannot ensure diversity, leading to bad exploration under deceptive environments. NSR-ES can converge fast, but will finally get trapped into a local optimum. This is because NSR-ES reproduces only one policy in each iteration, and may update the same policy for several iterations, which can cause a rapid improvement of a particular policy but may also lead to a sub-optimal policy due to the lack of considering the whole behavior space. Among the previous algorithms we have compared, only QD-RL can finally obtain the optimal policy, but its convergence speed is slow. We can observe that our proposed algorithm EDO-CS can converge to the optimal policy much faster, showing the advantage of clustering-based selection.

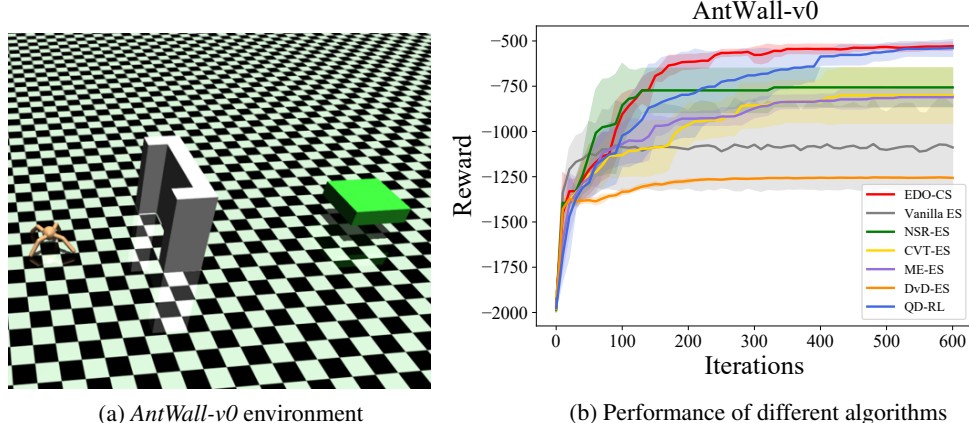

(a) *AntWall-v0* environment

(b) Performance of different algorithms

Figure 2: The *AntWall-v0* environment and the performance of different algorithms.

## 4.2 MULTI-MODAL ENVIRONMENTS

Next, we create two multi-modal environments to explicitly examine whether EDO-CS can efficiently find high-quality policies with diverse behaviors. The created *HalfCheetahFwdBwd* and *AntFwdBwd* environments are based on *HalfCheetah-v2* and *Ant-v2*, respectively. In the *HalfCheetahFwdBwd* environment, instead of rewarding the agent for walking forward only, we assign rewards for both walking forward and backward, implying that there are two modals in this environment, i.e., walking forward and walking backward. We train policies in this multi-modal environment, but test them in the single-modal environments *HalfCheetahFwd* and *HalfCheetahBwd*. The best-performing policies for these two modals are reported, respectively. We use the final location $(x, y)$ of the agent as the behavior characterization. The *AntFwdBwd* environment is created similarly.

To compare different algorithms more clearly, we organize the experimental results into a tabular form. Table 2 reports the cumulative rewards, together with the average performance rankings of all algorithms in the four environments. We can observe that Vanilla ES which maintains one policy can only learn a single modal, i.e., walking backward in *HalfCheetahFwdBwd* and walking forward in *AntFwdBwd*. As expected, all the other algorithms maintain a population of policies, and thus can achieve better performance than Vanilla ES in the multi-modal environments. DvD-ES can learn both modals in *AntFwdBwd*, but a single modal in *HalfCheetahFwdBwd*. NSR-ES, ME-ES, CVT-ES and QD-RL can learn both modals in both environments, but the quality of each modal is less satisfactory due to their inefficient selection mechanisms. We can observe that EDO-CS can learn any modal, and always achieves the highest quality in each modal, validating that EDO-CS can find a set of high-quality policies with diverse behaviors.

Table 2: Rewards obtained by different algorithms in multi-modal environments.

| Environment | EDO-CS | QD-RL | ME-ES | DvD-ES | CVT-ES | NSR-ES | Vanilla ES |
|---|---|---|---|---|---|---|---|
| *HalfCheetahFwd* | **4284** | 2930 | 2700 | -3419 | 3219 | 1346 | -5543 |
| *HalfCheetahBwd* | **6548** | 6013 | 5953 | 6353 | 4672 | 5366 | 3911 |
| *AntFwd* | **4617** | 4291 | 4316 | 4507 | 3856 | 1737 | 1911 |
| *AntBwd* | **4697** | 4164 | 4123 | 3498 | 2958 | 3961 | -851 |
| Performance Ranking | **1** | 3 | 3.5 | 3.75 | 4.75 | 5.25 | 6.75 |

### 4.3 SINGLE-MODAL ENVIRONMENTS

To examine the performance of EDO-CS for solving tasks that do not require much diversity, we conduct experiments in standard MuJoCo environments. Here we adopt the action-based behavior characterization, which was also used in (Parker-Holder et al., 2020). The behavior characterization $b(\pi_\theta)$ of a particular deterministic policy $\pi_\theta$ is represented in a vectorized form: $b(\pi_\theta) = \{\pi_{\theta(\cdot|s)}\}_{s \in \mathcal{S}}$, where $\mathcal{S}$ is the state space, and $\pi_\theta(\cdot \mid s)$ corresponds to the action taken in state $s$ under policy $\pi_\theta$. Since the state space can be infinitely large, we randomly sample 20 states instead of all. We compare all algorithms used before, except ME-ES, which requires discretization of the behavior space and is hard to be applied to high dimensional behavior space. Figure 3 shows the results of different algorithms on four standard MuJoCo tasks. We can observe that EDO-CS can always achieve the best final performance. Although Vanilla ES and NSR-ES can sometimes converge faster at the early stage of optimization, e.g., in the *Humanoid-v2* environment, their final performance is limited. The performance of DVD-ES and QD-RL is dominated by that of EDO-CS, because their curves are always below that of EDO-CS.

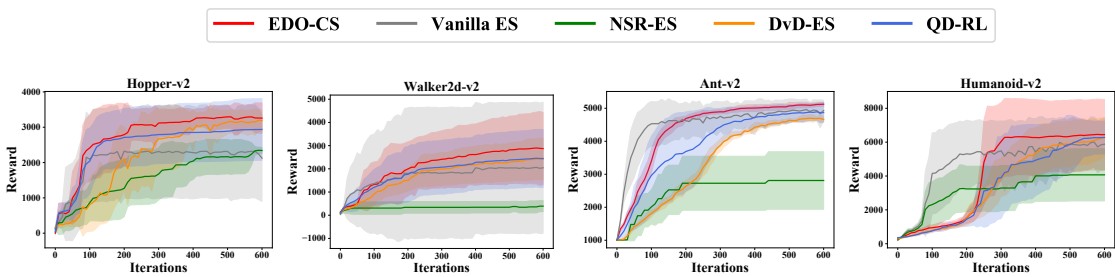

Figure 3: Performance of different algorithms in four standard MuJoCo environments.

By visualizing the clusters in different iterations of EDO-CS, we find that a set of diverse policies can be indeed selected from the archive through clustering-based selection. By the comparison with fixed $\lambda$, we also validate the effectiveness of self-adjusting $\lambda$ in Eq. (2) using multi-armed bandit modeling. The influence of the hyper-parameters (including clustering algorithm, number $T'$ of updating iterations, population size $K$ and archive size $l$) on the performance of EDO-CS is investigated as well. Due to space limitation, these experimental results are provided in Appendix A.2.

## 5 CONCLUSION

QD algorithms aim to help RL to find a set of high-quality policies with diverse behaviors in complex environments. However, existing algorithms cannot fully guarantee both quality and diversity, mainly because of their inefficient selection mechanisms. In this paper, we propose the algorithm EDO-CS using clustering-based selection, and demonstrate its superiority by experiments on various control tasks.

We have used ES as the optimizer. In fact, EDO-CS can also be equipped with the sample-efficient policy optimizer (Liu et al., 2020; Nilsson & Cully, 2021), which may further improve the performance. The mechanism of self-adjusting $\lambda$ by multi-armed bandit modeling can be replaced by state-of-the-art online-learning algorithms, which may better balance the importance of quality and diversity. In addition, how to decide the population size automatically is also a matter of interest. One possible approach is to let the clustering algorithm determine the number of clusters by itself. Other interesting future works include combining EDO-CS with representation or manifold learning (Vassiliades & Mouret, 2018; Gaier et al., 2020; Rakicevic et al., 2021) in the parameter space or behavior space, and analyzing the performance of EDO-CS theoretically (Gao et al., 2015; Doncieux et al., 2019).

ACKNOWLEDGMENTS

We thank the anonymous reviewers for their insightful and valuable comments. We thank Feiyu Liu, Lei Yuan, Ziniu Li, Tian Xu, Haopu Shang, and Xueyao Zhang for reading the manuscript and providing helpful comments. This work was supported by the NSFC (62022039), the Jiangsu NSF (BK20201247), and the CAAI-Huawei MindSpore Open Fund.

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

# A APPENDIX

## A.1 HYPER-PARAMETERS

This section will introduce the hyper-parameters used in our experiments. For fair comparison, all the hyper-parameters of different algorithms are set to be same in each environment.

Table 3: Hyper-parameters for deceptive and multi-modal environments.

| Hyper-parameter | AntWall-v0 | HalfCheetahFwdBwd | AntFwdBwd |
|---|---|---|---|
| Population size $M$ | 5 | 5 | 5 |
| Archive size $l$ | 20 | 100 | 100 |
| Number $T'$ of updating iterations | 10 | 5 | 10 |
| $\sigma$ in ES | 0.025 | 0.03 | 0.025 |
| $\eta$ in ES | 0.015 | 0.02 | 0.015 |

Table 4: Hyper-parameters for single-modal environments.

| Hyper-parameter | Hopper-v2 | Walker2d-v2 | Ant-v2 | Humanoid-v2 |
|---|---|---|---|---|
| Population size $M$ | 5 | 5 | 5 | 5 |
| Archive size $l$ | 30 | 30 | 30 | 20 |
| Number $T'$ of updating iterations | 5 | 5 | 10 | 10 |
| $\sigma$ in ES | 0.025 | 0.025 | 0.025 | 0.0075 |
| $\eta$ in ES | 0.02 | 0.03 | 0.015 | 0.02 |

## A.2 ADDITIONAL EXPERIMENTAL RESULTS

**Visualization of clusters.** We visualize the clusters in different iterations of EDO-CS for the experiments under the *AntWall-v0* environment. Figure 4 shows the behavior space, where the points represent the solutions in the current archive, and the different colors represent the different clusters. Throughout the optimization process, we can observe that the solutions in the archive can always be partitioned into clusters with different behaviors, validating the effectiveness of clustering-based selection. As the optimization goes, the clusters can spread different regions of the behavior space better, also implying that EDO-CS can indeed find a set of diverse policies.

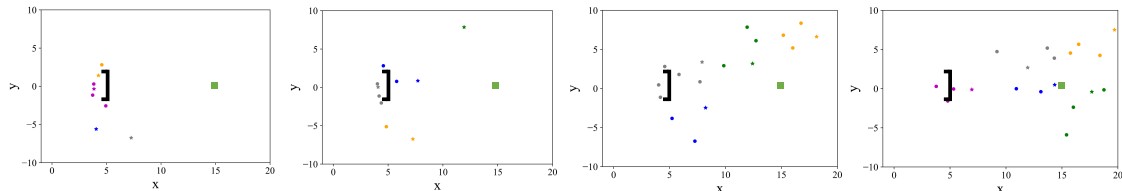

Figure 4: Visualization of the clusters in different iterations of EDO-CS under the *AntWall-v0* environment.

**Adaptive $\lambda$.** EDO-CS self-adjusts the hyper-parameter $\lambda$ in Eq. (2) by multi-armed bandit modeling. A larger $\lambda$ implies a larger weight of diversity when doing reproduction with ES. We have used $\{\lambda^{(1)} =$

$0, \lambda^{(2)} = 0.5$} as the two arms in the experiments. To examine the effectiveness of this adaptive mechanism, we compare it with fixed $\lambda \in \{0, 0.5\}$ in deceptive and multi-modal environments. The results in Table 5 show that the adaptive setting of $\lambda$ always leads to the best performance.

Table 5: Performance of EDO-CS under different settings of $\lambda$.

| Setting of $\lambda$ | AntWall-v0 | HalfCheetahFwd | HalfCheetahBwd | AntFwd | AntBwd |
|---|---|---|---|---|---|
| Adaptive $\lambda$ | **-529** | **4284** | **6548** | **4617** | **4697** |
| $\lambda = 0$ | -650 | 3856 | 5931 | 4340 | 4426 |
| $\lambda = 0.5$ | -850 | 3877 | 6077 | 2800 | 3093 |

**Direct optimization.** The selection process in each iteration of EDO-CS is to select a set of high-quality policies with diverse behaviors from the archive. One direct way is to optimize the weighted sum of the quality (i.e., rewards) and an explicit diversity metric, which is, however, quite difficult. First, it is hard to balance the weights of quality and diversity in the objective function. Second, this is actually a pseudo-Boolean optimization problem with the search space size $\binom{l}{K}$, where $l$ is the archive size and $K$ is the number of selected policies. Besides, this optimization problem requires an efficient optimization algorithm, which is, however, not easy to design. Here, we empirically compare our clustering-based selection strategy with the direct-optimization-based selection strategy.

For the direct optimization procedure, a set of high-quality policies with diverse behaviors is selected from the archive in each iteration, by directly optimizing the weighted sum of quality and diversity. The quality is defined as the sum of rewards of the selected policies. We consider two diversity measures. One is the sum of the pairwise distances between all the policies in the selected set. The other is the determinant of the similarity matrix $\Pi$ of the policies in the selected set, where $\Pi_{i,j}$ is the similarity between the $i$-th and $j$-th selected policies, and the main diagonal are all 1. The similarity matrix $\Pi$ is also used in DvD-ES (Parker-Holder et al., 2020). The two corresponding objective functions are denoted as $f_{pair}$ and $f_{det}$, respectively, as shown in Eqs. (8) and (9).

$$f_{pair} = (1 - \omega) \sum_{i=1}^{K} \mathbb{E}_{\tau \sim \pi_{\boldsymbol{\theta}_i}} [R(\tau)] + \omega \sum_{i=1}^{K} \sum_{j \neq i} \|b(\pi_{\boldsymbol{\theta}_i}) - b(\pi_{\boldsymbol{\theta}_j})\|_2 \tag{8}$$

$$f_{det} = (1 - \omega) \sum_{i=1}^{K} \mathbb{E}_{\tau \sim \pi_{\boldsymbol{\theta}_i}} [R(\tau)] + \omega \cdot det(\Pi) \tag{9}$$

The weight $\omega$ in Eqs. (8) and (9) is set to $1/2$ in the experiments, i.e., the quality and diversity are treated equally important. Note that both the quality and diversity have been normalized to be in $[0, 1]$. For the optimizer, we employ a genetic algorithm which maintains 100 solutions (i.e., the population size is 100) and generates 100 offspring solutions in each generation. To optimize $f_{pair}$ and $f_{det}$, we run the genetic algorithm for 200 generations.

The results are shown in the following two tables. EDO-CS is the proposed algorithm with the clustering-based selection strategy, while EDO-DOS$_{pair}$ and EDO-DOS$_{det}$ denote the algorithms with the direct-optimization-based selection strategy, where the objective functions are $f_{pair}$ and $f_{det}$, respectively. Table 6 shows the reward of the best policy found by each algorithm, from which we can observe that the EDO-CS algorithm performs better in most cases. Table 7 shows the time required to run the selection process once, and the clustering-based selection strategy is significantly faster than the direct-optimization-based selection strategy. Thus, our clustering-based selection method can be better and faster.

Table 6: Performance of EDO-CS and the algorithms with the direct-optimization-based selection strategy.

| Method | AntWall-v0 | HalfCheetahFwd | HalfCheetahBwd | AntFwd | AntBwd |
|---|---|---|---|---|---|
| EDO-CS | **-529** | **4284** | **6548** | 4617 | **4697** |
| EDO-DOS$_{pair}$ | -706 | 4188 | 5847 | **5013** | 4392 |
| EDO-DOS$_{det}$ | -536 | -5591 | 6529 | -530 | 2417 |

Table 7: Time required to run the selection process once of EDO-CS and the algorithms with the direct-optimization-based selection strategy.

| Method | AntWall-v0 | HalfCheetahFwdBwd | AntFwdBwd |
|---|---|---|---|
| EDO-CS | **0.015** | **0.021** | **0.023** |
| EDO-DOS$_{pair}$ | 9.570 | 9.802 | 9.807 |
| EDO-DOS$_{det}$ | 8.900 | 9.266 | 9.567 |

**Clustering algorithm.** An essential step of EDO-CS is to cluster policies in the behavior space. We have adopted the $K$-means clustering algorithm because of its popularity. However, it is interesting to examine whether the choice of clustering algorithm will affect the performance of EDO-CS largely. Thus, we conduct experiments over various clustering algorithms, including Hierarchical Agglomerative Clustering (HAC) (Murtagh & Legendre, 2014), $K$-means (Lloyd, 1982), Balanced Iterative Reducing Clustering using Hierarchies (BIRCH) (Zhang et al., 1996), and Spectral Clustering (Von Luxburg, 2007). Figure 5 shows that these clustering algorithms except BIRCH can lead to the similar performance of EDO-CS in most cases. BIRCH is suitable for a large number of samples, while the number of policies in the archive is relatively small. Thus, we may not get good clustering results by BIRCH, degrading the performance.

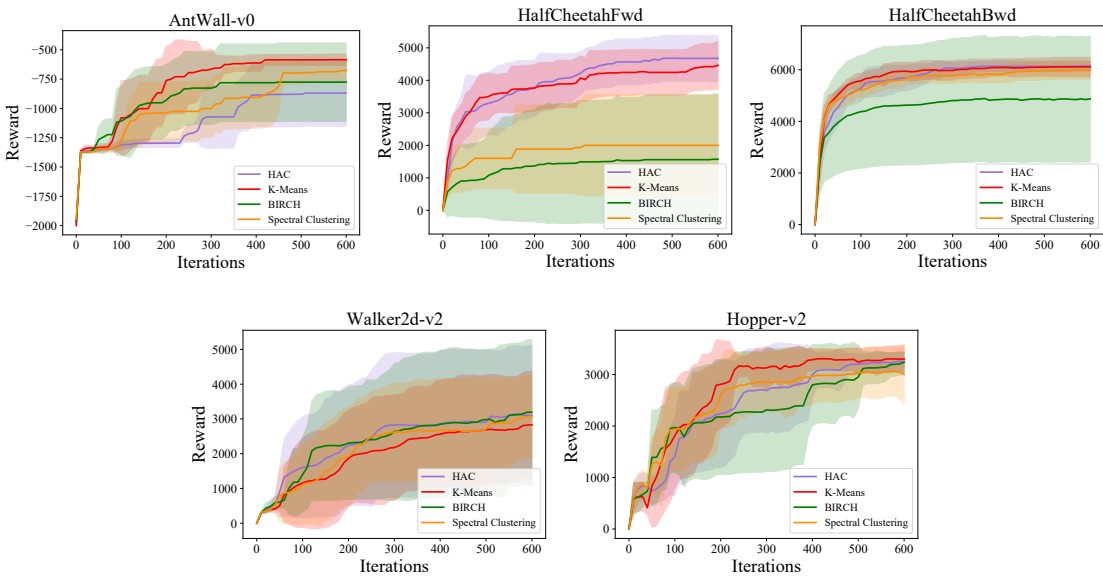

Figure 5: Performance of EDO-CS equipped with different clustering algorithms.

**Number $T'$ of updating iterations.**   We also study the influence of the number $T'$ of updating iterations, i.e., the number of iterations for which we use ES to update the policy (in lines 12–15 of Algorithm 1) after selecting it from the archive. We have tested $T' \in \{5, 10, 15, 20\}$ under various environments. The results in Table 8 show that the performance of EDO-CS is not very sensitive to $T'$. Note that we have also tried to update the selected policy for only one iteration (i.e., set $T' = 1$), but it fails. If the number of updating iterations is too small, the difference between solutions will be insignificant, leading to ineffective selection.

Table 8: Performance of EDO-CS under different numbers $T'$ of updating iterations.

| Environment | $T' = 5$ | $T' = 10$ | $T' = 15$ | $T' = 20$ |
|---|---|---|---|---|
| *HalfCheetahFwd* | 4462 | 4216 | 4911 | **4926** |
| *HalfCheetahBwd* | 6407 | **6527** | 6285 | 6203 |
| *Walker2d-v2* | 2830 | **4870** | 4362 | 4676 |
| *Hopper-v2* | 3301 | 3566 | **3578** | 3559 |

**Population size $K$.**   The population size $K$ refers to the number of policies we select from the archive in each iteration. If the population size is too small, we may not be able to find all optimal policies with diverse behaviors. If the population size is too large, some policies with similar behavior will be selected and updated, harming the efficiency of EDO-CS. Thus, it is important to select a proper population size $K$.

Table 9 shows the performance of EDO-CS under different population sizes. For a sample-wise fair comparison, we report the results under the same number of interactions with the environment rather than under the same number of iterations. It can be observed that the default population size of $K = 5$ is appropriate in most environments.

Table 9: Rewards obtained by EDO-CS with different population sizes $K$, given the same number of interactions with the environment.

| Environment | $K = 3$ | $K = 5$ | $K = 7$ | $K = 9$ |
|---|---|---|---|---|
| *AntWall-v0* | -852 | **-618** | -1082 | -970 |
| *HalfCheetahFwdBwd* | 9101 | **10274** | 10221 | 10242 |
| *Walker2d-v2* | 1840 | 2453 | 3062 | **3558** |
| *Hopper-v2* | 2688 | **3164** | 2781 | 2856 |

Figure 6 shows the performance of EDO-CS with different population sizes, given the same number of iterations. As expected, a larger population size implies a greater number of interactions with the environment, and thus results in a better performance of EDO-CS.

**Archive size $l$.**   The archive is used to record the high-quality policies with diverse behaviors generated-so-far. It is crucial for EDO-CS, as the parent policies for reproduction in each iteration are selected from the archive. If the archive size $l$ is too small, some useful information may be lost. If $l$ is too large, the archive will contain lots of low-quality policies encountered before, the selection of which for reproduction will lower the efficiency. Figure 7 shows the results under different archive sizes. It would be interesting to investigate how to adaptively set the archive size of QD algorithms.

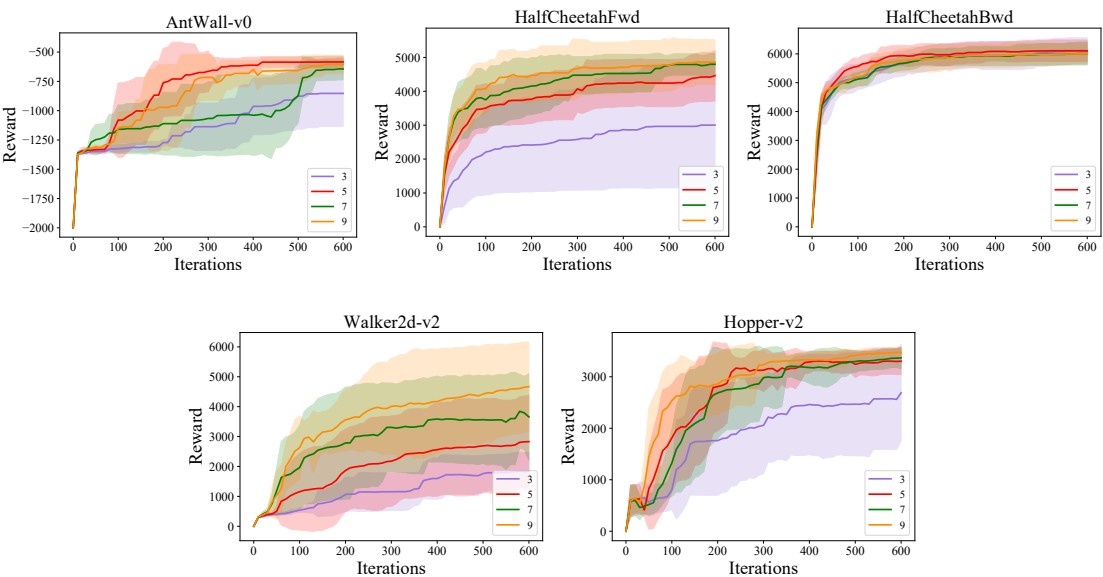

Figure 6: Performance of EDO-CS with different population sizes $K$, given the same number of iterations.

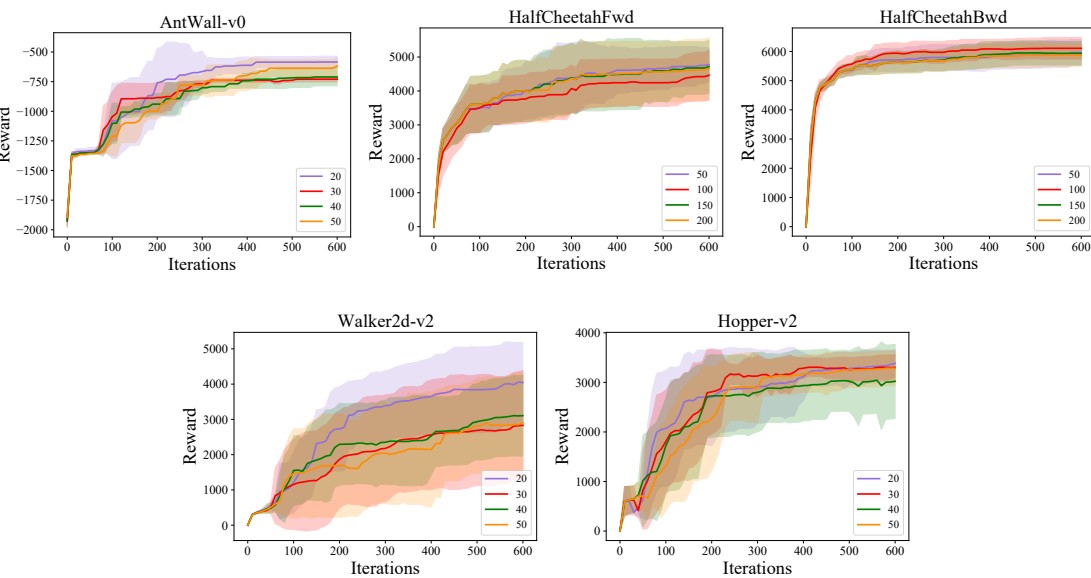

Figure 7: Performance of EDO-CS under different archive sizes.

