# OpenReview forum: "Evolutionary Diversity Optimization with Clustering-based Selection for Reinforcement Learning"
_ICLR.cc/2022/Conference — ICLR 2022 Poster_

### Official Review · Reviewer_SvUL · 2021-10-25

**Correctness:** 4
**Technical Novelty And Significance:** 3
**Empirical Novelty And Significance:** 3
**Recommendation:** 6
**Confidence:** 4

**Main Review:**


Strengths:
* Experiments are quite comprehensive, having performed comparisons over many different diversity algorithms. The results show that EDO-CS achieves better performance over all other baselines. The environments used are also quite diverse, ranging from AntWall, to multi-modal environments, and finally single-modal environments.
* Multiple ablation studies are performed over the method, including varying components such as the clustering algorithm used, number of ES update iterations, population size, and archive size. This should give practitioners better idea of how to tune the method.


Weaknesses:

* Some presentation issues throughout the paper.
  * In Figure 1(a), the dots should be made bigger and colors should be more visible and contrasting, or there should be a way to denote the region highlighted by e.g. blue or red dots. This will allow the conclusion to be much more obvious that Pareto/Clustering-selected points allow more multiobjective dominance / higher rewards, as currently it is fairly straining to the eyes. Similar issue occurs in Figure 1(b), where I had trouble understanding what is going on. What does "quality" (the heatmap metric) mean here? What should I be looking at? It may help to define these terms like "quality" first, which means it might be best to move Figure 1 after notation has been established in Section 2.
  * Could you explain why we're using a bandit approach for $\lambda$? If I'm not mistaken, normally multi-arm bandit scenarios involve a stochastic reward function defined by a probability distribution. Can you explain, what is the "reward function" for $\lambda$? If it's $J(\theta)$, isn't $\theta$ moving and thus "$\lambda$'s reward is nonstationary?" I'm fairly confused about the involvement of bandits in this scenario.
  * Slightly awkward English that could also be cleaner and more precise with the use of mathematical notation. For instance, Section 3.2: "Then, we select one specific policy from each cluster. More specifically, from the cluster where the best quality policy locates, we just select this policy." This sentence could be made more precise by using the notation defined previously (e.g. cluster $C_{i}$, policy $\pi_{\theta_{i}}$). This was an issue throughout the paper, where there were too many ambiguous references.
  * A less important issue is in formatting, where there is a gap in space at the bottom of each page that should be fixed. Grammar, spacing, and \citep vs \cite issues should be fixed throughout.

* Somewhat shaky motivation for the clustering heuristic.
  * The real end goal of the clustering procedure, is to really find $K$ policies which are behaviorally different from each other. The rest of the clustering doesn't ever actually get used (please correct me if I'm wrong). If this is the case, why is clustering the best way to find $K$ diverse policies? Normally, clustering algorithms use an entire cluster for some purpose (e.g. see applications in [1]). Why not optimize an explicit diversity metric that measures the behavioral difference between the $K$ policies? I admit, I have seen a variant of the clustering approach (termed speciation) in NEAT [2], but it would significantly strengthen this paper if there was a better motivation + explanation for the use of the clustering heuristic.
  * Another way to explain this better is to perhaps visualize the clusters in terms of behaviors, throughout optimization, on a 2-d plot (since some of the Mujoco benchmarks conveniently define $b(\cdot)$ in terms of xy-locations).


[1] https://en.wikipedia.org/wiki/K-means_clustering

[2] Evolving Neural Networks Through Augmenting Topologies (2002)

**Summary Of The Paper:**

This paper performs research in the area of diversity/novelty-seeking agents under evolutionary optimization approaches. The core idea in this area is to optimize a weighted combination between raw environment reward and diversity during training (even though evaluation is still on the raw environment reward only). The diversity acts as a "regularizer" in order to achieve better exploration, which can lead to higher rewards.

The specific approach (EDO-CS) by the paper suggests using a pool/"archive" $A$ of previous policies, which are then clustered via K-means according to the distance metric defined by the behavior function $b(\cdot)$. Each cluster then produces a new policy (and thus there are a total of $K$ new policies), which are then updated via ES over a regularized reward (via diversity metric, along with an adaptive weighting $\lambda$), and put back into the archive.

Experiments are performed over various continuous control benchmarks for testing exploration, and results show that EDO-CS achieves SOTA.

**Summary Of The Review:**

While the paper produces experimentally strong results, it lacks motivation (especially for the clustering heuristic) as well as certain components such as the use of multi-armed bandits for adaptively tuning $\lambda$. One danger to this lack of motivation is the introduction to potential confounding factors (e.g. what if the clustering heuristic is implicitly optimizing an ideal diversity objective, and it's better to optimize this ideal diversity objective instead?).

There is also a nontrivial amount of presentation issues (not just small grammar mistakes or typos) that affect the paper's scientific quality as well.

I am willing to increase my score if the authors fixed the core issues outlined above.

EDIT: I have increased my score to a 6 as well as my confidence scores. The authors have addressed my questions and concerns.

---

> ### Author Response · Authors · 2021-11-21
> **Response to Reviewer SvUL**
>
> Thank you for your constructive comments.
>
> [Q1] Some issues of Figure 1.
>
> [R1] Thanks to your suggestion, we have revised Figure 1 and the corresponding explanation.
>
> [Q2] Could you explain why we're using a bandit approach for $\lambda$?
>
> [R2] The objective function in Eq.(3) is equal to $(1-\lambda) \cdot reward +\lambda\cdot diversity$, where the reward and diversity correspond to exploitation and exploration, respectively. Thus, the parameter $\lambda$ controls the trade-off between exploitation and exploration. As the trade-off can vary from task to task as well as at different stages of the same task, we want to design a strategy to automatically adjust the value of $\lambda$. Thus, we introduce an online mechanism for balancing exploitation and exploration adaptively, i.e., we model it as a multi-arm bandit problem and adopt the UCB algorithm to select $\lambda$.
>
> [Q3] What is the reward function of the bandit?
>
> [R3] For each arm (i.e., each value of $\lambda$), the gain of rewards after updating a policy with this $\lambda$ is used as its reward. We want to clarify that to use the multi-arm bandit model, we have made a stationary assumption, i.e., disregarding the change of the reward distribution of each arm in the process of optimization. In the future, it would be interesting to design a more fine-grained reward function for the bandit.
>
> [Q4] The writing and the use of mathematical notation.
>
> [R4] Thanks for your suggestion. We have added more mathematical notations in the updated version.
>
> [Q5] The issues of formatting.
>
> [R5] Thanks for your suggestion. We have fixed the grammar and formatting issues as you suggested. However, a gap in space at the bottom of each page still exists, which is required by the format of ICLR.
>
> [Q6] Motivation for the clustering heuristic.
>
> [R6] The selection process in each iteration of EDO-CS is to select a set of high-quality policies with diverse behaviors from the archive. As you indicated, one direct way is to optimize the weighted sum of the quality (i.e., rewards) and an explicit diversity metric, which is, however, quite difficult. First, it is hard to balance the weights of quality and diversity in the objective function. Second, this is actually a pseudo-Boolean optimization problem with the search space size $C^k_n$, and requires an efficient optimization algorithm, which is, however, not easy to design. Thus, we employ the clustering-based selection process: clustering the policies in the archive and then selecting a high-quality policy from each cluster, which maybe not the best way, but is natural and efficient. We have revised to add some explanation. Thank you.
>
> [Q7] Another way to explain this better is to perhaps visualize the clusters in terms of behaviors.
>
> [R7] Thanks for your suggestion. We have revised to visualize the clusters in different iterations of EDO-CS for the experiments under the *AntWall-v0* environment. As shown in the Appendix A.2 of the revised version, the clusters can well represent different regions of the behavior space, thus validating the effectiveness of clustering-based selection.

---

> > ### Comment · Reviewer_SvUL · 2021-11-21
> > **Increased my score**
> >
> > Thanks for the clarifications and update to the draft. The draft seems to be much cleaner now. I've updated my score to a 6.
> >
> > I want to suggest to the authors especially about answering [Q6], where I think it's very important to include response [R6] in the paper and provide its motivation and comparison against a direct optimization procedure over a diversity objective.

---

> > > ### Author Response · Authors · 2021-11-22
> > > **Response to reviewer SvUL**
> > >
> > > Thanks for your suggestion. We will include it into the final version. We will try to run the emprical comparison with a direct optimization procedure, and report the results during the rebuttal phase.

---

> > > > ### Author Response · Authors · 2021-11-29
> > > > **Response to reviewer SvUL (experiment)**
> > > >
> > > > Dear Reviewer SvUL, thanks to your suggestion, we have empirically compared our clustering-based selection strategy with the selection strategy based on direct optimization.
> > > >
> > > > For the suggested direct optimization procedure, a set of $K$ high-quality policies with diverse behaviors is selected from the archive in each iteration, by directly optimizing the weighted sum of quality and diversity. The quality is defined as the sum of rewards of the selected policies. We consider two diversity measures. One is the sum of the pairwise distances between all the policies in the selected set. The other is the determinant of the similarity matrix of the policies in the selected set, which is also used in DvD-ES.
> > > >
> > > > The two corresponding objective functions are denoted as $f_{pair}$ and $f_{det}$, respectively, as shown in Eqs. (1) and (2). The weight $\omega$ is set to $1/2$ in the experiments, i.e., the quality and diversity are treated equally important. Note that both the quality and diversity have been normalized to be in $[0,1]$.
> > > > $$
> > > > f_{pair}=(1-\omega)\sum_{i=1}^K r(\pi_{\boldsymbol\theta_i}) + \omega \sum_{i=1}^K\sum_{j \neq i}d(b_{\pi_{\boldsymbol\theta_i}},b_{\pi_{\boldsymbol\theta_j}}) \ \ (1)
> > > > $$
> > > >
> > > > $$
> > > > f_{det} = (1-\omega)\sum_{i=1}^K r(\pi_{\boldsymbol\theta_i}) + \omega \det(\Pi)\ \ (2)
> > > > $$
> > > >
> > > > For the optimizer, we employ a genetic algorithm which maintains $100$ solutions (i.e., the population size is $100$) and generates $100$ offspring solutions in each generation. To optimize $f_{pair}$ or $f_{det}$, we run the genetic algorithm for $200$ generations.
> > > >
> > > > The results are shown in the following tables. EDO-CS is the proposed algorithm with the clustering-based selection strategy, while `optimization_pair` and `optimization_det` denote the algorithms with the direct-optimization-based selection strategy, where the objective functions are  $f_{pair}$ and $f_{det}$, respectively.
> > > >
> > > > | Method            | AntWall-v0 | HalfCheetahFwd | HalfCheetahBwd | AntFwd   | AntBwd   |
> > > > | :---------------- | ---------- | -------------- | -------------- | -------- | -------- |
> > > > | EDO-CS            | **-529**   | **4284**       | **6548**       | 4617     | **4697** |
> > > > | optimization_pair | -706       | 4188           | 5847           | **5013** | 4392     |
> > > > | optimization_det  | -536       | -5591          | 6529           | -530     | 2417     |
> > > >
> > > > | Method            | AntWall-v0 | HalfCheetahFwdBwd | AntFwdBwd |
> > > > | :---------------- | ---------- | ----------------- | --------- |
> > > > | EDO-CS            | 0.015      | 0.021             | 0.023     |
> > > > | optimization_pair | 9.570      | 9.802             | 9.807     |
> > > > | optimization_det  | 8.900      | 9.266             | 9.567     |
> > > >
> > > > The first table shows the reward of the best policy found by each algorithm, from which we can observe that the EDO-CS algorithm performs better in most cases. The second table shows the time required to run the selection process once, and the clustering-based selection strategy is significantly faster than the direct-optimization-based selection strategy. Thus, our clustering-based selection method can be better and faster.
> > > >
> > > > We will include this comparison in the final version. Thank you.

---

> > > > > ### Comment · Reviewer_SvUL · 2021-11-29
> > > > > **Thank you for the comparisons**
> > > > >
> > > > > Hi, thank you for the relevant comparisons against an explicit diversity objective. This is very interesting, as it seems like the clustering method (which isn't explicitly optimizing any diversity objective) still outperforms explicit diversity optimization. I guess this suggests that there might be some sort of implicit effect on the trajectory/loss landscape when using the clustering-based method which allows better performance.
> > > > >
> > > > > I haven't seen this type of result before (previously I also had reservations about the use of clustering/'speciation' in NEAT, but I'm more convinced now that it is doing something significant), and thus I've learned something new as well. I will update my confidence/novelty scores accordingly.

---

### Official Review · Reviewer_5sru · 2021-11-02

**Correctness:** 3
**Technical Novelty And Significance:** 1
**Empirical Novelty And Significance:** 2
**Recommendation:** 5
**Confidence:** 5

**Main Review:**

The experimental evaluation demonstrates the benefits of the proposed methods on a variety of domains showing different numbers of modalities and complexity of the behavioural space. Multiple existing algorithms are used as baselines, which is great.

Overall the paper is very clear and easy to read and understand. It is well structured and nicely illustrated.

Unfortunately, while it is a great paper, it misses some key references in the literature which are making the novelty claims less obvious and inviting additional experiments.

First of all, the concept of dividing the behaviour space into cells and only selecting the best solutions is exactly the purpose of the grid-based archive in MAP-Elites algorithms[1], probably the most famous QD algorithm in the literature. Combining MAP-Elites with a K-mean to automatically partition the behaviour space was firstly introduced in CVT-MAP-Elites [2] and its extension Cluster-Elites [3]. Biasing the cell/cluster selection based on the reward/fitness was first studied in [4] (without really demonstrating improvement over a uniform selection, when quality and diversity of the archive is considered).
Moreover, the comparison between selection from the archive, or from a separate population or a Pareto-front, was also studied in [4] reaching the same conclusion as in the proposed paper: archive-based selection outperforms population or Pareto-front ones.

It is important to come back on the fact that the grid of MAP-Elites already performs a form of clustering and filtering to keep only the best policies in each cell of the grid. Given that ME-ES (which stand for MAP-Elites-ES) uses such a grid, I found the statement "only selecting one policy in each iteration [used by ME-ES], may cause only a specific behaviour to be well optimized and can not return a set of high-quality policies with diverse behaviours." confusing as the grid is designed to maintain a diversity of behaviour and improving them in parallel.

In my opinion, the proposed selection mechanism is not novel and directly linked to the archive management introduced in MAP-Elites and its extensions.

The paper should also consider PGA-MAP-Elites[5], which combines CVT-MAP-Elites with policy gradient (TD3) for the evolution of NN policies and outperforming the best baseline (QD-RL) presented in this paper. Given that PGA-MAP-Elites uses a selection mechanism that is linked to the one proposed in this paper and shows better performance than QD-RL and other baselines, I think it is an important baseline to consider.

Additionally, it seems that the authors of the QD-RL paper are still trying to publish the paper and made several updates to the paper. https://openreview.net/pdf?id=8FRw857AYba
One of them is the use of the grid from MAP-Elites. This change was most likely made because the authors found similar findings to the one presented here. I do not think it is fair to request the authors of this paper to be aware and to consider a paper that is still under review somewhere else. Therefore, I am not considering this part in my assessment of the paper. Yet, I thought it is important to point the authors to this new version of the paper so that the claims made here will not be outdated in a few months.

Other, smaller comments:
The use of the best reward makes sense from a deep-RL point of view. However, the sum of the reward of all the policies in the collection (also, called QD-score) is more frequently adopted in the QD literature. It would be good to add a few sentences justifying this decision and how it can affect the baselines' performance.
In the multi-modal environment, how is computed the Mean rewards? Is it the mean reward of the two best policies (one per modality), or something else?
"Figure 6: Archive size" does not seem to report archive size.

[1] Mouret, J. B., & Clune, J. (2015). Illuminating search spaces by mapping elites. arXiv preprint arXiv:1504.04909.

[2] Vassiliades, V., Chatzilygeroudis, K., & Mouret, J. B. (2017). Using centroidal voronoi tessellations to scale up the multidimensional archive of phenotypic elites algorithm. IEEE Transactions on Evolutionary Computation, 22(4), 623-630.

[3] Vassiliades, V., Chatzilygeroudis, K., & Mouret, J. B. (2017, July). A comparison of illumination algorithms in unbounded spaces. In Proceedings of the Genetic and Evolutionary Computation Conference Companion (pp. 1578-1581).

[4] Cully, A., & Demiris, Y. (2017). Quality and diversity optimization: A unifying modular framework. IEEE Transactions on Evolutionary Computation, 22(2), 245-259.

[5] Nilsson, O., & Cully, A. (2021). Policy gradient assisted MAP-Elites. In Proceedings of the Genetic and Evolutionary Computation Conference Companion.

**Summary Of The Paper:**

In the paper "Evolutionary Diversity Optimization With Clustering-Based Selection For Reinforcement Learning", the authors introduce a new selection mechanism for Quality-Diversity based algorithms. This new selection mechanism is based on the K-Means algorithm, to cluster the behaviour space into cells and then select only the best policies within each cell. The paper also uses a linear combination of the reward and novelty score when performing the policy updates. The weights the of the linear combination are automatically adjusted using a bandit algorithm so that the best reward is obtained.

**Summary Of The Review:**

Overall the paper is very clear and easy to read and understand. It is well structured and nicely illustrated.
Unfortunately, while it is a great paper, it misses some key references in the literature which are making the novelty claims less obvious and inviting additional experiments.

----
Updated score after discussion below.
----

---

> ### Author Response · Authors · 2021-11-21
> **Response to Reviewer 5sru (1/2)**
>
> Thank you for your constructive comments.
>
> [Q1] The concept of dividing the behavior space into cells and only selecting the best solutions is exactly the purpose of the grid-based archive in MAP-Elites algorithms[1], probably the most famous QD algorithm in the literature.  Combining MAP-Elites with a K-means to automatically partition the behavior space was firstly introduced in CVT-MAP-Elites [2] and its extension Cluster-Elites [3]. Biasing the cell/cluster selection based on the reward/fitness was first studied in [4] (without really demonstrating improvement over a uniform selection, when quality and diversity of the archive is considered).
>
> [R1] According to [4],  there are two key components for QD algorithms, i.e., container and selection operator, as shown in the following table. Although both the MAP-Elites methods [1,2,3] and our approach EDO-CS use clustering to ensure diversity, their usages are quite different, which focus on the management of the container and the selection of solutions for reproduction, respectively.
>
> | Container      | Selection Operators                                          |
> | -------------- | ------------------------------------------------------------ |
> | Grid / Archive | No selection / Uniform selection / Score proportionate selection / Population-based selection / Pareto-based selection |
>
> The MAP-Elites methods [1,2,3] use the grid-based container, and employ clustering to better partition the behavior space. Particularly, clustering can be employed to partition a high-dimensional space into well-spread geometric regions.
>
> Our approach EDO-CS employs clustering to address the limitation of existing selection operators. By clustering the solutions (i.e., policies) in the archive according to their behavior distance and selecting a high-quality solution from each cluster, a set of high-quality solutions with diverse behaviors can be selected for reproduction. Thus, EDO-CS introduces a new selection operator for QD.
>
> Accordingly, there are many differences of using clustering. For example, clustering will be used only once in CVT-MAP-Elites [2], which is performed to partition the behavior space before the iterative evolutionary process; while it will be used for selection in each iteration of the proposed EDO-CS. Furthermore, the number of clusters is quite different. EDO-CS uses five clusters in our experiments, while to partition the entire behavior space well, CVT-MAP-Elites [2] often requires a large number of clusters, e.g., 5000 or 10000.
>
> Thanks to your suggestion, we have revised to discuss the relation and difference at the end of Section 3.2. We have also tried to add these methods [1,2,3] in the empirical comparison. As EDO-CS uses ES to update the policy, ES also needs to be employed in these methods [1,2,3] for a fair comparison. MAP-Elites-ES [1] is similar to ME-ES, which we compared in the original paper. The results of CVT-MAP-Elites-ES [2] are shown in the following table, from which we can observe that EDO-CS is still better. We have not implemented Cluster-Elites-ES, because its details are not clearly presented in [3].
>
> | Method            | AntWall-v0 | HalfCheetahFwd | HalfCheetahBwd | AntFwd   | AntBwd   |
> | ----------------- | ---------- | -------------- | -------------- | -------- | -------- |
> | CVT-MAP-Elites-ES | -801       | 3219           | 4672           | 3856     | 2958     |
> | EDO-CS            | **-529**   | **4284**       | **6548**       | **4617** | **4697** |
>
> In fact, these two usages of clustering can be combined to achieve a better performance, because they focus on two different components of the QD algorithms, i.e., container management and selection operator. For example, for some high-dimensional tasks, it is hard to use clustering to get a good partition of the behavior space, and thus the uniform selection operator may fail to select a set of high-quality solutions with diverse behaviors; while the proposed clustering-based selection operator can help address this issue. This would be an interesting future work.

---

> > ### Comment · Reviewer_5sru · 2021-11-29
> > **Thank you for your response.**
> >
> > [Q1]
> > While I understand that managing the archive and selecting from it is two different elements of a QD algorithm, the distinction between the two following approaches is quite subtle:
> > 1) clustering the space in cells and keeping only the best solution found in each cell and then performing a uniform selection over the cells, and
> > 2) Keeping all the solutions, and then clustering the space and selecting only the best solution in each cluster.
> >
> > However, the fact that the number of clusters is significantly smaller than what we usually observe in the literature is interesting. My understanding is that by shifting the clustering from the archive management to the selector with fewer cells EDO-CS creates a stronger selective pressure without potentially losing some stepping stones. The additional experiment with CVT-MAP-Elites-ES is an interesting step in this direction.
> >
> > However, it is inaccurate to say: "Previous works use the grid-based container and employ clustering to partition a high-dimensional behavior space into well-spread geometric regions before running the iterative evolutionary process, while our EDO-CS employs clustering to select a set of high-quality solutions with diverse behaviors from the archive in each iteration." as Vassiliades et al., 2017; 2018 (cited in the sentence preceding the quote) use respectively the Voronoi diagram from k-mean and update the corresponding centroids at each generation. There are differences between these methods but not the one described in the paper.
> >
> > The difference between these different methods should be more explicit in the paper and more deeply studied as it remains subtle, but lead to promising results based on the latest addition to the paper. However, it remains unclear where the performance difference comes from, whether it is the smaller number of clusters, the periodic update or the varying metrics.
> >
> > [Q2]
> > The paper [4] not only considers a Pareto-based selection from a population (this is one of the 19 variants studied in the paper), but also selection operators based on the archive and which use different biases (uniform, fitness, novelty, or curiosity). These experiments showed that the selection operator can play an important role in the performance.
> >
> > [Q3]
> > Great.
> >
> > [Q4]
> > Yes, that makes sense.
> >
> > I believe that the manuscript has improved with the additional experiment and the reference to the literature. Therefore, I have updated my score to 5. However, there are still some analyses and comparisons with the literature that would help improve the paper even further.

---

> > > ### Author Response · Authors · 2021-11-30
> > > **Response to Reviewer 5sru 2**
> > >
> > > Thanks for pointing out the inaccurate description about [Vassiliades et al., 2017; 2018] and the
> > > paper [4] in [Q2]. We will revise to make them precise in the final version.
> > >
> > > To make fair comparisons in the experiments, we have set the hyper-parameters of the algorithms as same as possible, e.g., using the same number of generations for running ES in each iteration of the algorithms. Thus, we believe that the advantage of the proposed EDO-CS over CVT-MAP-Elites-ES is mainly due to the MAB selection of $\lambda$ (i.e., the trade-off parameter between the quality and diversity), and the clustering-based selection strategy. As suggested by Reviewer oa8b, we have added an ablation study to compare EDO-CS with fixed $\lambda \in \{0,0.5\}$ and adaptive $\lambda$ by MAB selection. The results shown in Table 5 of the revised version validate the effectiveness of adaptive $\lambda$. By comparing EDO-CS with $\lambda= 0$ (i.e., the third row in Table 5) and CVT-MAP-Elites-ES (i.e., the second row in the table of [R1]), we find that EDO-CS is better. This discloses the benefit of the clustering-based selection strategy, because the other components of EDO-CS with $\lambda= 0$ and CVT-MAP-Elites-ES are the same. We next explain why the clustering-based selection strategy (employed by EDO-CS) can be better than the strategy of uniform selection over the cells (employed by CVT-MAP-Elites-ES).
> > >
> > > When using the QD algorithm for solving RL tasks, selection is crucial due to the limited computational resources. In each iteration, we need to select a set of high-quality solutions with diverse behaviors to update, which is, however, difficult for the algorithms that cluster the entire behavior space and perform uniform selection, such as CVT-MAP-Elites-ES. When the number of clusters (i.e., cells) is large, the solutions worthy of updating may not be selected using uniform selection; when the number of clusters is small, some potential solutions (i.e., stepping stones) will be discarded. In contrast, EDO-CS maintains an archive with diverse solutions, and further performs clustering for selection (i.e., clusters the diverse solutions in the archive according to their behavior distance and selects a high-quality solution from each cluster). That is, CVT-MAP-Elites-ES only uses clustering to generate a set of diverse cells and then performs uniform selection, while besides maintaining a diverse archive, EDO-CS further uses clustering for selection. Thus, as you indicated, EDO-CS can create a stronger selective pressure without potentially losing some stepping stones, which guarantees selecting a set of high-quality solutions with diverse behaviors for reproduction. If we replace the uniform selection strategy of CVT-MAP-Elites-ES with the clustering-based selection strategy, i.e., CVT-MAP-Elites-ES first clusters all the solutions in the cells and then selects the best solution from each cluster, the performance is expected to be improved. We will add this experiment as well as these analyses and discussions in the final version.
> > >
> > > Finally, we want to thank you and the other reviewers again for providing these constructive comments and helping us improve the work.

---

> ### Author Response · Authors · 2021-11-21
> **Response to Reviewer 5sru (2/2)**
>
>
> [Q2] Conclusion of this paper is same in [4]: archive-based selection outperforms population or Pareto-front ones.
>
> [R2] The Pareto-based selection strategy in [4] selects solutions from the population (instead of the archive) via Pareto ranking, which can be seen as an extension of the population-based selection strategy. Thus, [4] mainly shows that archive-based selection outperforms population-based one, which is consistent with our experimental results, e.g., the algorithms (i.e., EDO-CS, QD-RL and ME-ES) using archive-based selection are better than those (i.e., DvD-ES, NSR-ES and Vanilla ES) using population-based selection in Table 2. But more importantly, our results show that different archive-based selection strategies will have a large influence on the performance. For example, the proposed algorithm EDO-CS using clustering-based selection performs better than the previous algorithm QD-RL using Pareto-based selection. Note that the Pareto-based selection strategy used in QD-RL selects solutions from the archive, different from that mentioned in [4], which selects solutions from the population.
>
> [Q3] Description of ME-ES in the paper.
>
> [R3] Thanks for pointing out this. We are sorry that we did not explain it clearly. We have revised it in the updated version.
>
> [Q4] Should also consider PGA-MAP-Elites [5]?
>
> [R4] Because we mainly focus on the selection operator, we keep the optimizer consistent, i.e., all baselines use ES as the optimizer. PGA-MAP-Elites combines CVT-MAP-Elites with a gradient-based optimizer. For fairness comparison, we need to replace the gradient-based optimizer in PGA-MAP-Elites with an ES optimizer. The modified PGA-MAP-Elites is exactly CVT-MAP-Elites-ES, which we have revised to compare according to your suggestions in [Q1]. The experimental results in [R1] show that the proposed EDO-CS performs better than CVT-MAP-Elites-ES. It is worth noting that all the compared algorithms can be combined with the gradient-based optimizer, and we leave this for future work.
>
> [Q5] New version of QD-RL, the claims made here maybe outdated.
>
> [R5] Thanks for your notice. We have checked the new version of QD-RL, and found that the authors mainly added a new algorithm QD-TD3+ME. One main contribution of our work is indicating the drawback of Pareto-based selection of QD-RL, and then proposing the new clustering-based selection strategy. However, their new version has not realized this point. As in the conclusion of their paper, they claim that "We could replace the maximization of the sum of reward contributions with a multi-criteria selection from a Pareto front, where diversity would be only one of the considered criteria.", i.e., they are still promoting Pareto-based selection. Actually, their goal of adding QD-TD3+ME is mainly to show that their "QD+TD3" approach can be equipped with different QD algorithms, such as MAP-Elites. Their experimental results show that QD-TD3+ME is better sometimes while QD-TD3+Pareto is better in other cases, without providing any explanation. Thus, we think that our claim is not affected by this new version. Thanks to your suggestion, we will continue to track the work of QD-RL.
>
> [Q6] Some smaller comments.
>
> [R6] Here are the responses for the smaller comments.
>
> - We did not use QD-score, because there are more direct ways to compare the performance of the algorithms under our tested environments. In the environments with deceptive rewards, the goal is to find a policy which can bypass the obstacles efficiently. Thus, using the reward of the best policy (e.g., in Figure 2) is sufficient.  In the multi-modal environments, the goal is to find a high-quality policy for each modal. Thus, we compared the reward of the best policy for each modal (e.g., in Table 2). Thanks to your suggestion, we have revised to add some explanation.
>
> - Here ''Mean rewards'' refers to the average of the rewards we get from evaluating the best policy multiple times in the environment. We have revised to make it clear.
>
> - We have revised the caption of Figure 6 to "Performance of EDO-CS under different archive sizes". Thank you.

---

### Official Review · Reviewer_oa8b · 2021-11-03

**Correctness:** 3
**Technical Novelty And Significance:** 3
**Empirical Novelty And Significance:** 4
**Recommendation:** 6
**Confidence:** 4

**Main Review:**


The paper is well-written with a clear goal developing a method that optimises for diversity in addition to fitness. The main contributions include sampling policies and a novel approach to adapting the optimisation function.

However, I have several concerns:

* The definition of a behaviour is not clear from the beginning and should be defined earlier. There seems to be a misalignment between the use of behaviour diversity in the proposed approach (mostly for exploration) w.r.t. the approaches in the literature (to maintain a diverse collection of policy behaviours). It would be useful to discuss how the proposed approach relates/differs from the two main approaches in QD:

  - Joel Lehman and Kenneth O Stanley. Evolving a diversity of virtual creatures through novelty search and
local competition, GECCO, 2011
  - Jean-Baptiste Mouret and Jeff Clune. 2015. Illuminating search spaces by mapping elites. ArXiv, 2015
  - Antoine Cully, Jeff Clune, Danesh Tarapore, and Jean-Baptiste Mouret. Robots that can adapt like animals. Nature, 2015

* If I understand correctly, the clustering is done in the policy parameter space rather than in the behaviour space. This leads to the diversity in policy parameterisation but not necessarily to behaviour diversity. This then also raises a question what is the point of maintaining these clusters as opposed to sampling based on distance. Another drawback I see is that there is no explicit maintenance of diversity within the archive, and thus no way of sampling diverse behaviours.
On this note, it would be beneficial to provide some additional reasoning behind this. Moreover, there are several recent works that discuss using learned representations or manifolds in order to obtain a more grounded notion of distance in the policy parameter space, which might be useful in this work:
  - Nemanja Rakicevic, Antoine Cully, and Petar Kormushev. Policy manifold search: Exploring the manifold hypothesis for diversity-based neuroevolution. GECCO, 2021
  - Adam Gaier, Alexander Asteroth, and Jean-Baptiste Mouret. Discovering Representations for Black-box Optimization. GECCO, 2020
  - Vassilis Vassiliades and Jean-Baptiste Mouret. Discovering the elite hypervolume by leveraging interspecies
correlation. GECCO, 2018

* I think an ablation study to show the contribution of the MAB selection of  $\lambda$ would be beneficial, as this seems to be an important contribution. This ablation would further strengthen the point made about optimising for diversity.


### Some questions:
* What are the results that are reported, of the best performing policy in each iteration?

* The total number of iterations in the deceptive task is 600, does this mean the total number of policies evaluated in the environment?
Section 4.3 distinguishes number of interactions with the environment vs number of iterations. This is not clear from Algorithm 1, and would be useful to clarify. The presented number of evaluations is significantly less than what is usually needed for ES-based approaches.


### Minor comments:

* It is not clear what does "EAs type" in Table 1 mean exactly. In the text it says “how many parents and offspring the algorithm will maintain in the population” but I find it not clear what this means.



**Summary Of The Paper:**

The authors present EDO-CS an approach to multi-objective optimisation that ensures the high quality and diversity of generated RL policies. Instead of uniformly sampling solutions from the Pareto front, sampling policies from learned clusters is introduced, as a selection mechanism within the ES optimisation. The quality and diversity are further achieved by modifying the objective function of the ES algorithm so it includes both the fitness and behaviour-diversity terms, balanced with a hyperparameter $\lambda$. This hyperparameter is set using a multi-arm bandit approach, which is adapted during the training.
The performance of the proposed approach is evaluated on several MuJoCo continuous control tasks, as well as compared to state-of-the-art QD benchmarks.


**Summary Of The Review:**

- Interesting approach to maintaining the policy clusters and adapting the objective function
- Needs to improve the context in the literature, w.r.t. the use of diversity of behaviours
- An ablation study to show the contribution of the MAB for objective function adaptation, would be beneficial

EDIT: I have increased my Empirical Novelty And Significance score, and conditioned on the additional results added to the camera ready I would be willing to increase my recommendation.

---

> ### Author Response · Authors · 2021-11-21
> **Response to Reviewer oa8b (1/2)**
>
> Thank you for your constructive comments.
>
> [Q1] The definition of behaviour is not clear from the beginning and should be defined earlier.
>
> [R1] We use the coordinates of the final location $(x,y)$ of the robot as the behavior characterization in the deceptive and multi-modal environments, and use the actions taken by policies as the behavior characterization in the single-modal environments. Thanks to your suggestion, we have revised to define the behavior of a policy in Section 2, before introducing the proposed algorithm.
>
> [Q2] There seems to be a misalignment between the use of behavior diversity in the proposed approach with the approaches in the literature.
>
> [R2] In fact, we use the behavior diversity to achieve both of these two goals.
>
> - The first one is to explore better in the environments with deceptive rewards, as illustrated by our experiments in the *AntWall-v0* environment in Section 4.1. The experimental results show that our approach can enable the Ant to get over the obstacles and reach the goal efficiently.
> - The second goal is to obtain a set of high-performing policies with diverse behaviors, as illustrated by our experiments in the *HalfCheetahFwdBwd* and *AntFwdBwd* environments in Section 4.2. The experimental results show that our approach can find two optimal policies with diverse behaviors, i.e., walking forward and backward.
>
> [Q3] It would be useful to discuss how the proposed approach relates/differs from the two main approaches in QD.
>
> [R3] Relation: All these methods share a common evolutionary framework: maintaining an archive/population, and iteratively selecting parent solutions from the archive/population and updating them to reproduce offspring solutions.
>
> Difference:
>
> - Selection mechanism. NSLC [Lehman and Stanley, GECCO'11] uses Pareto-based selection (similar to our baseline QD-RL), MAP-Elites [Mouret and Clune, ArXiv'15; Cully et al., Nature'15] generally uses uniform selection (same as our newly added baseline CVT-MAP-Elites-ES), while we propose a new clustering-based selection mechanism. The experimental results show that our selection mechanism is superior.
> - Reproduction mechanism. Instead of using the GA operator (i.e., mutation) as NSLC and MAP-Elites, we use the ES optimizer, which is more suitable for high-dimensional optimization problems, e.g., in RL.
>
> Thanks to your suggestion, we have revised to discuss the difference between these two approaches and those QD approaches for RL in the second paragraph of Section 1.
>
> [Q4] The clustering is done in the policy parameter space rather than in the behavior space?
>
> [R4] We do clustering in the behavior space. For the standard formulation of K-means, i.e., Eq. (4) in the paper, each $\boldsymbol{x}$ actually corresponds to the behavior characterization $b(\pi_{\boldsymbol{\theta}})$ of a policy $\pi_{\boldsymbol{\theta}}$, as described in the paragraph following Eq. (4). We have revised to make it clear.
>
> [Q5] No explicit maintenance of diversity within the archive, and thus no way of sampling diverse behaviors?
>
> [R5] When updating the archive in line 18 of Algorithm 1, we have maintained its diversity by avoiding adding policies with similar behaviors. To be specific, whenever we want to add a policy into the archive, we first select the policy with the most similar behavior from the archive. If their behavior distance is greater than a threshold, we add the new policy into the archive. Otherwise, we compare their rewards and keep the policy with the higher rewards. We are very sorry that we did not describe it clearly in the original paper. Thanks to your comment, we now have revised to explain it in detail at the end of Section 3.1.
>
> [Q6] Several recent works that discuss using learned representations or manifolds, which might be useful in this work.
>
> [R6] Thanks for your suggestion. We have revised to add it in our future work.
>
> [Q7] An ablation study to show the contribution of the MAB selection of $\lambda$.
>
> [R7] Thanks for your suggestion. We have added this part in the Appendix A.2 of the revised version. We examine the effectiveness of adaptive $\lambda$ by comparing EDO-CS with fixed $\lambda \in \{ 0,0.5 \}$ and adaptive $\lambda$ in Eq.(3) in deceptive and multi-modal environments. The experimental results show that the adaptive $\lambda$ leads to the best performance.
>
> [Q8] What are the results that are reported, of the best performing policy in each iteration?
>
> [R8] For QD algorithms, the archive is usually returned as the final output result. Thus, we report the best-performing policy in the current archive in each iteration.

---

> ### Author Response · Authors · 2021-11-21
> **Response to Reviewer oa8b (2/2)**
>
> [Q9] The total number of iterations in the deceptive task is 600, does this mean the total number of policies evaluated in the environment?
>
> [R9] In each iteration, EDO-CS selects $K$ policies to update in parallel. So one iteration corresponds to $K$ generations of the ES optimizer. During one generation of the ES optimizer, it needs to evaluate several policies (the number of evaluated policies is denoted by $m$) in the environment to estimate the gradient. In our experiments, the population size $K$ is always set to 5. Thus, the total number of policies evaluated in the environment is $600 \times 5 \times m=3000m$. For example, $m=60$ in the *AntWall-v0* environment, and thus the total number of evaluations is 180000.
>
> [Q10] Section 4.3 distinguishes number of interactions with the environment vs number of iterations. This is not clear from Algorithm 1, and would be useful to clarify.
>
> [R10] In Section 4.3 of the original paper (which is moved into Appendix A.2 of the revised version), we want to study the effect of the population size $K$ on the performance of EDO-CS. In each iteration, EDO-CS selects $K$ policies to update in parallel. To have a sample-wise fair comparison, we keep $K\times T$ consistent, where $T$ refers to the total number of iterations. For example, if we report the results of 600 iterations under population size 5, then we report the results of 300 iterations under population size 10. We are very sorry that we did not describe it clearly in the original paper. Thanks to your comment, we now have explained it in detail in the revised version.
>
> [Q11] It is not clear what does "EAs type" in Table 1 mean exactly. In the text it says “how many parents and offspring the algorithm will maintain in the population” but I find it not clear what this means.
>
> [R11] In the column of "EAs type" in Table 1, we use $(\mu+\lambda)$ to denote that the algorithm maintains $\mu$ solutions in the population, and generates $\lambda$ offspring solutions in each iteration.  We have revised to make it clear. Thank you.

---

> > ### Comment · Reviewer_oa8b · 2021-12-01
> > **Thank you for your response.**
> >
> > I would like to thank the authors for taking the extra time and efforts to address my concerns, as well as those of other reviewers. I believe that this paper has improved during the review process. Although there are still some minor issues pointed out by other reviewers, I believe this paper would be a good resource to the QD-RL community.
> > I am going to increase the significance score, and if the results of the empirical comparisons with the selection strategy based on direct optimization would be added in the camera ready version I would be willing to increase my recommendation score as well.

---

> > > ### Author Response · Authors · 2021-12-01
> > > **Response to Reviewer oa8b 2**
> > >
> > > Dear reviewer oa8b, we will add the comparison with the selection strategy based on direct optimization into the final version. Thank you very much.

---

### Official Review · Reviewer_RfKF · 2021-11-30

**Correctness:** 4
**Technical Novelty And Significance:** 3
**Empirical Novelty And Significance:** 3
**Recommendation:** 8
**Confidence:** 3

**Main Review:**

The paper gives clear motivation to the problem of optimising for quality and diversity in reinforcement learning and clearly defines relevant terminology throughout. It is easy to follow and the conclusion are well supported by the results.

More specifically to the method presented, the authors investigate the applicability to clustering methods for selection of policies in evolutionary strategies. This is an interesting area of study and the authors present a convincing method that performs well on several toy problems. There has been some application of evolutionary strategies for selection of policies in the field but this is the first application of K-means clustering to my knowledge. One of my initial criticisms when reading was the impact of hyper-parameters (cluster size, clustering algorithm, archive size, etc) but was pleased to see that the authors have considered this in detail and presented relevant results on this within the appendices.

The central comment I have on this paper is it would be interesting to see how their method navigates between clusters and how representative these clusters are of behavioural strategies. Evolutionary algorithms are well known for falling into local minima given poor selection of mutation rates and population size. As I understand the paper, the clustering aspect addresses this by ensuring a wider range of policy-space is sampled. Is it possible to quantify/comment on whether the authors method is achieving this? Similarly, I naively assume that cluster separation will be arbitrary where the number of clusters is larger than the number of distinct behaviours. I would be interested to see how well the clustering algorithm performs at identifying behaviours – for example, a demonstration that policies sampled from the clusters do indeed lead to diverse behaviours when examining those policies that were ranked highly but distinct.

Finally, there are some minor recommended typographic edits. In several places, method initials are given without prior statement (for example, ME-NS and ES). The colour and size of points in Figure 1 is also not extremely clear, especially when trying to identify Pareto/Clustering/Both/None in Fig.1b.


**Summary Of The Paper:**

The authors present a new method for finding robust policies in reinforcement learning, those policies which give high rewards but also come from a diversity of behaviours. The authors accomplish this by using a clustering algorithm to first divide policies into relevant clusters and then use evolutionary algorithms to optimise policy choice. The result is an algorithm that performs well over several representative benchmarks, achieving SOTA performance with high convergence rates.

**Summary Of The Review:**

Overall, an interesting and well thought through investigation into the role of clustering when selecting in evolutionary strategies for reinforcement learning. The results give a clear indication that clustering can greatly improve results under certain settings, with good attention given to the impact on changes method parameters such as clustering algorithm and size. However, more needs to be done to show explicitly that those selected policies do end up being more diverse in behaviours.

---

### Author Response · Authors · 2021-11-21
**General response to reviewers**


We appreciate valuable comments from all reviewers. We have revised our paper carefully according to your suggestions. The modification parts are as follows.

- Overview
  - We revise the writing, grammar, mathematical notation and formatting throughout the paper.
  - We revise Figure 1 and the corresponding explanation to make it clearer.
  - We add figures to visualize the clusters in different iterations of EDO-CS in Appendix A.2.
  - We add a new baseline to the experiments.
  - We add an ablation study to validate the effectiveness of the multi-arm bandit selection of $\lambda$ in Appendix A.2.
  - We add some references.

- Introduction:
  - We revise to discuss the difference between the EDO-CS and the two classical QD algorithms in the second paragraph of Section 1.
  - We revise Figure 1 and the corresponding explanation in the sixth paragraph of Section 1.

- Background:
  - We define the behavior of a policy in Section 2, before introducing the proposed algorithm.

- The EDO-CS algorithm
  - We rewrite the description of the archive in Section 3.1.
  - We rewrite Section 3.2 to make the motivation of using clustering clearer.
  - We add the comparison between EDO-CS and some previous QD algorithms using clustering in the last paragraph of Section 3.2.
  - We explain the motivation and some details of the multi-arm bandit, including the reward function for $\lambda$ in Section 3.4.
  - We explain the meaning of “EAs type” in Section 3.5.

- Experiment
  - We add CVT-ES as a new baseline in Section 4.1 and 4.2.
  - We move the ablation study of population size $K$ to Appendix A.2.

- Conclusion
  - We add some future works to further improve EDO-CS.

- Appendix
  - We add figures to visualize the clusters in different iterations of EDO-CS for the experiments under the *AntWall-v0* environment in Appendix A.2.
  - We add an ablation study to validate the effectiveness of the multi-arm bandit selection of $\lambda$ in Appendix A.2.

- References
  - We add the following references in our paper.
    - A. Cully and J.-B. Mouret.  Behavioral repertoire learning in robotics. GECCO, 2013.
    - A. Cully and Y. Demiris. Quality and diversity optimization: A unifying modular framework. IEEE TEvC, 2018.
    - A. Gaier, A. Asteroth, and J.-B. Mouret. Discovering representations for black-box optimization. GECCO, 2020.
    - J.-B. Mouret and J. Clune. Illuminating search spaces by mapping elites. ArXiv, 2015.
    - N. Rakicevic, A. Cully, and P. Kormushev. Policy manifold search: Exploring the manifold hypothesis for diversity-based neuroevolution. GECCO, 2021.
    - V. Vassiliades and J.-B. Mouret. Discovering the elite hypervolume by leveraging interspecies correlation. GECCO, 2018.
    - V. Vassiliades, K. I. Chatzilygeroudis, and J.-B. Mouret. A comparison of illumination algorithms in unbounded spaces. GECCO Companion, 2017.
    - V. Vassiliades, K. I. Chatzilygeroudis, and J.-B. Mouret. Using centroidal voronoi tessellations to scale up the multidimensional archive of phenotypic elites algorithm. IEEE TEvC, 2018.

---

### Decision · Program_Chairs · 2022-01-20

**Decision:**

Accept (Poster)

**Comment:**

This paper introduces a novel quality-diversity algorithm, "Evolutionary Diversity Optimization with Clustering-based Selection (EDO-CS)", and applies it to reinforcement learning. A bandit approach (UCB) is used to select which cluster to sample parents from. The QD algorithm can be evaluated on its own, outside of the RL context, and if so it should be compared to the several approaches to niching and other standard diversity preservation approaches in evolutionary computation that rely on clustering. (And the authors should make an effort to connect to the niching literature in particular.) However, the use of the algorithm for RL makes it possible to use behavioral features as the space in which to cluster, separating it from standard diversity preservation methods. The resulting algorithm is relatively simple and the empirical results are good.

Some of the main concerns for reviewers included the bibliography, which the authors promptly acted on by citing several suggested papers and comparing their approach where relevant. There was also discussion about the exact novelty of the paper, for example as compared to the CVT-MAP-Elites algorithm, but this was clarified by the authors. Reviewers agree that the paper is easily to follow and well-written.

Based on this, it seems that the paper makes a clear contribution to QD methods for RL, and is worth accepting.